# Does Viewing Green Advertising Promote Sustainable Environmental Behavior? An Experimental Study of the Licensing Effect of Green Advertising

**Chenyu Gu** [1,2] [ID]**, Shiyu Liu** [1] **and Subai Chen** [1,2,*]

1 School of Journalism and Communication, Xiamen University, Xiamen 361005, China
2 Shenzhen Research Institute of Xiamen University, Xiamen University, Shenzhen 518057, China
* Correspondence: subaichenxmu@126.com or subai_chen@xmu.edu.com; Tel.: +86-182-5033-5337

**Abstract:** Current research on consumer behavior of green advertising mostly focuses on advertising attitude or consumer behavior, while few studies have extended the topic to explore the consumers' behavior after green consumption. The "Licensing effect", which is a paradoxical side effect of green advertising, has been verified to exist in the consumption context in many countries. This paradoxical effect between cognition and behavior refers to the circumstance that consumers show non-green behavior after green consumption, which is contrary to the original intention of green advertising. However, at present, few scholars have verified and deeply explored that effect in the context of China. This study explores the "licensing effect" of green advertising through two factors: environmental protection cognition and advertising appeal. Through a 2 × 3 experiment, we find that: 1. The licensing effect is applicable in the Chinese consumption context; 2. The licensing effect only exists in individuals with low environmental protection cognition; 3. The appeal mode of green advertising turns out to be an effective moderator, and rational appeal can effectively prevent the licensing effect. This research expands the research scope of green advertising and provides a new vision for the study of consumer behavior in green advertising. In addition, the moderation role of advertising appeal verified by our study has guiding significance for green advertising practice.

**Keywords:** licensing effect; green advertising; green marketing; advertising appeal; environmental awareness; consumer behavior

## 1. Introduction

Sustainable development is of vital interest to every human being. Due to the perceived changes in the global natural environment and the media's publicity on environmental problems, both companies and consumers are paying increasing attention to environmental issues [1]. This has brought about an explosion in green consumption, with one study showing that global green advertising budgets have increased nearly tenfold in the last 20 years [2]. However, some scholars point out that the theoretical research related to green advertising has been carried out slowly in the field of advertising, especially the research from the audience's perspective [3]. Therefore, there is a need to advance the research from this perspective. Many studies around the world have shown that consumers in general exhibit positive attitudes toward green advertising [4,5]. However, most of the studies stop at consumers' green advertising attitudes and green consumer behavior, which cannot be fully equated with green behavior. Sustainable development emphasizes not only "development", but also "sustainability" as its important premise. The ultimate goal of green advertising research should be to explore whether green advertising can really influence the audience and make them develop sustainable green behavior habits. In response, some research has shown that green advertising does not always lead to sustainable green effects and that consumers may be less likely to follow up with green behavior after viewing green ads for green consumption, a paradox of inconsistent behavior

known as the "licensing effect" [6]. The licensing effect of green advertising runs counter to the underlying logic of the "cognitive congruence effect" in psychology. However, there are still some studies that have shown the opposite of the licensing effect, that is, green consumption leads to more follow-up green behavior by consumers [7].

Both effects are undeniably valid in green consumption, but the factors that lead to different behavioral outcomes remain to be discussed. Previous studies have shown that different approaches to green advertising and different personal traits have an impact on individual behavior [8,9]. Based on this, the current study introduces two factors, "individual environmental cognitive" and "green advertising appeal", and by carrying out a $2 \times 3$ experimental study we find that: when the type of green advertising is rational green appeal and non-green appeal, there is no license effect; when the type of green advertising is an emotional appeal, only individuals with low environmental cognitive level have a licensing effect. The findings also explain the seemingly contradictory existence of both the licensing effect and the cognitive congruence effect for green advertising; in other words, the licensing effect is also a manifestation of cognitive congruence for individuals with low environmental cognition.

The licensing effect has been proven by researchers in many countries (e.g., the United States, France, Canada, the Netherlands, etc.) [10]. Yet few scholars have conducted empirical studies on it in China, which is the world's most populous country; therefore, the green behavior of Chinese people is of great significance for environmental protection. The significance of the current study is that: first, it complements the applicability of the licensing effect in China and verifies that it is a universal mechanism applicable to global consumers; second, it expands the scope of research on green advertising by explaining the logic of licensing effect through research analyzing advertising content from an audience perspective; finally, this study exposes the moderating role of advertising appeal methods, which can provide a practical reference for preventing the occurrence of licensing effect.

## 2. Theoretical Framework

### 2.1. Green Advertising

Green advertising emerged from the late 1980s to the early 1990s, after scholars had discussed more environmental advertising [11]. Banerjee and other scholars have outlined three criteria for green advertising: first, the content of the advertisement explicitly or implicitly indicates the relationship between the product or service and the ecological and physical environment; second, the advertisement advocates environmental protection and a green lifestyle; third, the advertisement presents an image of the advertiser taking social responsibility for environmental protection; satisfying one of these criteria can be considered as a green advertisement [12]. In contrast, non-green advertising, which is the other side of the coin, refers to advertising that emphasizes attributes other than those of ecological conservation [13]. Green advertising has not only become a popular form of advertising in the industry but also attracted intense research interest among researchers in academia.

Existing research on green advertising mainly focuses on the direction of the product, advertising content, media platform, and audience. For example, Atkinson et al. argued that the eco-labeling of the product source, the novelty of the argument, and the degree of product involvement affect consumer trust, which in turn affects the effectiveness of green appeals [14]. Chang et al. studied the message framing of green advertising content based on the level of explanation theory and argued that the benefit framing positively influences the audience's willingness to buy green when matched with a high level of explanation [15]. Cao et al. studied green stream advertising on social media platforms and found that consumers prefer green advertising with higher social attributes [16]. From the perspective of the audience's self-construction, Mao et al. verified through two experiments that the emotional appeal of green advertising has more influence on the purchase intention of independent self-type audiences, while the rational appeal has more influence on interdependent self-type audiences [17]. Another perspective of green advertising research is to compare it with non-green advertising in order to explore why

green advertising is superior. One reason consumers buy green advertising products is that this behavior is often seen as environmentally ethical, and by doing so, people complete their self-identification as good people [18]. Another reason is that people consider the purchase of green advertising products as a symbol of reaction to their own environmental awareness and tend to associate this behavior with their personal social reputation [19].

There is no doubt about the "advertising" effect of green advertising, which has already been confirmed by a lot of studies, but does the "green effect " of green advertising only serve the "advertising"? Can the "green effect" still work after the audience has completed the purchase behavior? At present, the focus of green advertising research is rarely on the final benefits of green advertising, that is, consumers reduce their non-environmental consumption behaviors after they watch the advertisements, which is contrary to our original intention of advocating for "green advertising". Although some existing studies show that after consumers have purchased green products once, the possibility of their subsequent environmental protection behavior will increase [20], some studies in recent years have found the opposite phenomenon, with some consumers exhibiting contradictory behavior which is not environmentally friendly after making a green purchase [21], which is called the "licensing effect".

### 2.2. Licensing Effect, Cognitive Congruence Theory and Advertising Appeals

The licensing effect refers to the phenomenon that individuals who engage in ethical behavior will allow themselves to reduce ethical choices in the future or even increase unethical behavior [22,23]. Green behavior is often seen as ethical behavior; therefore, a number of scholars have researched and confirmed the existence of the licensing effect in the field of green behavior and green consumption. One study found that when people exhibit green behavior, they are significantly less likely to donate to environmental charities afterward [24]. Longoni and Gollwitzer et al. conducted an online experiment in which they asked one group of subjects to simulate an online green consumption scenario and another control group to simulate non-green consumption and then asked the subjects to perform a paper-cutting test to test people's green behavior by counting the subjects' sorting and disposal of leftover paper-cutting waste. The result showed that the green consumption group behaves significantly less green than the non-green consumption group [25]. It has also been found that even if an individual only expresses his attitude of supporting green consumption, it could also lead to a contradiction between his green attitude and actual behavior, thus causing the licensing effect. [26]. Accordingly, this study poses the following research questions:

RQ1: Is there a licensing effect on the effectiveness of green advertising in China?

There are also findings that contradict the licensing effect. People will maintain ethical behavior and green consumption consistently [27,28]. For example, some studies have shown that people are more inclined to demonstrate pro-social behavior when they purchase charitable products [29]. These studies are based on the underlying logic of cognitive congruence theory. Cognitive congruence theory, proposed by Leon Festinger, suggests that cognitive dissonance occurs when individuals are confronted with new information that contradicts their self-perceptions or perform behaviors that are contrary to their perceptions; people tend to avoid inconsistent behaviors in order to avoid this dissonance and maintain consistency in their inner world [30]. In terms of apparent behavior, the licensing effect is contradictory to the cognitive congruence theory, as the former shows inconsistency in pre- and post-behavior, while the latter states that individuals tend to avoid inconsistency in pre- and post-behavior; however, when examined internally there is no contradiction between the two from a cognitive perspective. The cognitive congruence theory is formulated with respect to cognition as two ways in which individuals will reduce the sense of dissonance from cognitive incongruence, either by adding new cognitions to reduce that dissonance or by avoiding increasing the degree of cognitive dissonance by avoiding exposure to new cognitions. From a cognitive perspective, the licensing effect may be a manifestation of cognitive dissonance, and it has been shown that environmental cognition is an important

antecedent of green behavior [31]. Thus, from the context of green consumption, those individuals with a low level of environmental protection cognition triggered cognitive dissonance in themselves after engaging in green consumption behaviors and thus generated subsequent avoidance of green behaviors to avoid increasing the degree of this cognitive dissonance; in contrast, those individuals with a high level of environmental protection cognition did not generate this sense of cognitive dissonance. Accordingly, we propose the following hypothesis:

**Hypothesis 1.** *The level of personal environmental protection cognition is positively correlated with green behavior.*

**Hypothesis 2.** *The licensing effect existed in the green advertising effect, but only in the group with low environmental protection cognition level.*

Based on the above logic, cognition is a key factor in generating the licensing effect, so the impact of green advertising on audience cognition should also be taken into consideration in the study. Previous studies have shown that different advertising appeals can have different effects on audience cognition, attitude and behavior [32,33]. Generally speaking, advertising appeal methods can be divided into two types: rational appeal and emotional appeal. Scholars have conducted a comparative analysis of the two types of appeals, and Laskey et al. argued that rational appeal advertising is superior to emotional appeal advertising in terms of explaining the message of the product's functional attributes, because rational appeal advertising is simpler and clearer than emotional appeal advertising in conveying key product information [34]. Aker's study also found that rational appeal advertising received higher scores in terms of "effectiveness" than rational appeal advertising [35]; Chan's study concluded the opposite, that audiences rated emotional appeal advertising more positively [36]. It is too one-sided to simply evaluate which advertising method is better: rational appeal or emotional appeal, the appeal of the advertising should match the product and the context. In the context of green advertising, the benefit claims brought by green consumption are different from traditional appeals. Compared with traditional appeals, audiences generally have relatively little knowledge of green appeal, so it is unwise to blindly adopt the advertising method of emotional appeal [37]. Therefore, we suppose that the use of rational appeal is more likely to achieve the effect of advertising persuasion. In other words, green advertising with rational appeal is less likely to have a licensing effect. Based on this, the following hypotheses are proposed in this study:

**Hypothesis 3.** *There is a licensing effect for the effect of green advertising with emotional appeal.*

**Hypothesis 4.** *There is no licensing effect for the effect of green advertising with rational appeal.*

**Hypothesis 5.** *The effect of green advertising with rational appeal on the audience's green behavior is more significant compared with green advertising with emotional appeal.*

### 3. Materials and Methods

In this study, a 2 × 3 inter-group experimental design was adopted to explore the relationship between the cognitive level of environmental protection (low vs. high) and the type of advertisement, which entails non-green advertisement, emotional appeal green advertisement, and rational appeal green advertisement. The green advertisement used in our experiment does not involve the specific actual brand, because the specific brand information may have an impact on consumers' attitudes, thus affecting the accuracy of the experimental results [38]. We use green advertising materials to achieve the purpose of environmental protection by calling on people to buy products made of environmentally friendly materials. The specific advertisements used are shown in Appendix A Figures A1–A3.

*3.1. Grouping of Environmental Protection Cognition Levels*

Since the research objective of this experiment is to investigate the licensing effect in groups of different environmental protection cognition levels, in order to effectively

differentiate the environmental protection cognition level of the sample, 214 volunteers were recruited through social media and their environmental protection cognition level was measured through a questionnaire before the formal experiment began. Since eight participants dropped out of the test midway, 206 valid questionnaires were finally obtained. The environmental protection cognition level scale was designed by drawing from Sparks and Whitmarsh [39,40]. Since the original version of the scale is in English, we invited two English masters to translate the scale in both directions and to conduct a pretest to ensure that the subjects could accurately and clearly understand the meaning of the measurement questions. The formal scale includes five questions: "1, I am a person who cares about environmental protection issues." 2, "I value myself as an environmentally friendly person." 2, "It is important to me that others regard me as an environmentally friendly person." 4, I prefer to buy green products compared to non-green products." 5, "I try to make my behavior environmentally friendly in my daily life." The scale was standardized on a 7-point Likert scale from 1 (completely disagree) to 7 (completely agree) with Cronbach $\alpha = 0.874$. The sample was then divided into high and low groups using the median as the boundary, and 90 samples were selected from low to high in the low group as group A (low environmental protection cognition level), and 90 samples were selected from high to low in the high group as group B (high environmental protection cognition level). The remaining 26 samples withdrew from the experiment and received a $1 reward.

The measurement results showed that the level of environmental protection cognition in group A was significantly lower than that in group B (MA = 3.704, SDA = 0.318; MB = 5.533, SDB = 0.504; $p < 0.001$). Therefore, we consider the distinction of environmental protection cognition level to be valid. The 180 volunteers were identified as the final subjects with an age distribution of 19–35 years old and an average age of 24.2 years old, 82 males accounted for 45.6% and 98 females accounted for 54.4%, specifically shown in Table 1. All participants were given a $3 reward after completing the experiment. Group A was then randomly divided into three groups with low environmental protection cognition levels and group B was randomly divided into three groups with high environmental protection cognition as shown in Table 2; each group was assigned to watch advertisements with different appeal types.

**Table 1.** Statistical table of basic information of effective samples.

| Statistical Items | Specific Content | Statistical Value | Percentage |
|---|---|---|---|
| Gender | Male | 82 | 45.6% |
| | Female | 98 | 54.4% |
| Age | 19–26 | 102 | 56.7% |
| | 27–40 | 63 | 35.0% |
| | Over 40 | 15 | 8.3% |
| Educational Background | High School and below | 19 | 10.5% |
| | Undergraduate | 84 | 46.7% |
| | Master and Doctor | 77 | 42.8% |

**Table 2.** Distribution of Subjects in each Condition Group of the Experiment (*n* = 180).

| NGA | | EAGA | | RAGA | |
|---|---|---|---|---|---|
| Low | High | Low | High | Low | High |
| Group 1 | Group 2 | Group 3 | Group 4 | Group 5 | Group 6 |
| 30 | 30 | 30 | 30 | 30 | 30 |

NGA = non-green advertising, EAGA = emotional appeal green advertising, RAGA = rational appeal green advertising, Low = low environmental protection cognition level, High = high environmental protection cognition level. The unit of measurement is the number of people.

### 3.2. Advertising Appeal Types as Variable

In this study, three kinds of advertisements were designed for different groups: non-green advertisements, emotionally appealing green advertisements and rationally appealing green advertisements. The three types of ads use exactly the same typography and background images, distinguished by the advertising copy. According to the literature

review and the actual situation, sports shoes are a common category of personal consumer goods, and as they are used by people with different demographic characteristics, sports shoe advertising was chosen to be the stimulus [41]. In order to avoid the influence of irrelevant factors and the subjects' existing impressions of the product brand, the product logo was concealed from the experimental images, and the shoe models were chosen for both men and women. The text of the non-green advertisement emphasized the function and comfort of the sports shoes, the emotional appeal green advertisement emphasized the environmental protection material of the sports shoes through emotional text, and the rational appeal green advertisement described the specific environmental protection material of the sports shoes and the specific contribution to environmental protection. Referring to previous studies on the common initiation of the permission effect [42], subjects in each group were asked to imagine that they had purchased sports shoes in the advertisement after viewing it.

### 3.3. Environmental Behavioral Intention

We then measured the environmental behavioral intention of all subjects and compared them between groups. The Environmental Behavioral Intention Scale was derived from Minton and Rose's study and adapted to the Chinese consumption context [43], finally containing five measures: (1) I am willing to sign a petition to support environmental protection; (2) I am willing to pay higher prices for green products; (3) I am willing to stop buying products from companies that do not act in an environmentally friendly way, even though it may inconvenience me; (4) I am willing to make an effort for the environment, even though the immediate results may not be significant; (5) I am willing to boycott companies that cause pollution to the environment. The scale was standardized on a 7-point Likert scale from 1 (completely disagree) to 7 (completely agree) with Cronbach $\alpha = 0.853$.

## 4. Results

### 4.1. Environmental Protection Cognition × Advertising Appeal Types (Emotional Appeal Green Advertising vs. Non-Green Advertising)

The results of data analysis showed that the main effect of personal environmental protection cognition was significant, ($F(1, 116) = 355.004$, $p = 0.00$, Partial $\eta^2 = 0.754$) while the main effect of advertising appeal types (emotional appeal green advertising vs. non-green advertising) was not significant, ($F(1, 116) = 0.148$, $p = 0.701$, Partial $\eta^2 = 0.001$). There was a significant interaction between environmental protection cognition and advertising appeal types on environmental behavioral intention ($F(1, 116) = 11.976$, $p = 0.001$, Partial $\eta^2 = 0.094$), as shown in Figure 1.

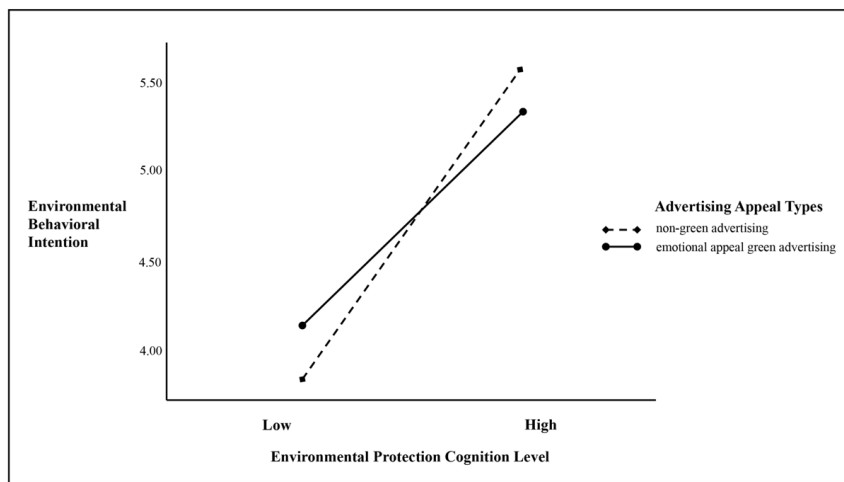

**Figure 1.** Emotional Appeal Green Advertising vs. Non-green Advertising.

Further comparative analysis showed that for the low environmental protection cognition level group, personal environmental behavioral intention after viewing green advertising was significantly lower than that of the non-green advertising viewing group ($M_{green}$ = 3.827, $SD_{green}$ = 0.266; $M_{non}$ = 4.127, $SD_{non}$ = 0.447; F = 6.693, $p$ = 0.003, 95% CI = [−0.4902; −0.1098]); for the high environmental protection cognition level group, personal environmental behavioral intention after viewing green advertising was not significantly different from the personal environmental behavioral intention after viewing non-green advertising group ($M_{green}$ = 5.567, $SD_{green}$ = 0.458; $M_{non}$ = 5.327, $SD_{non}$ = 0.499; F = 0.116, $p$ = 0.057, 95% CI = [−0.0077; −0.4877]).

In summary, we can assume that the licensing effect exists only in the low environmental protection cognition level group who view green advertising. In other words, the low-level environmental protection cognition group viewing green advertising (emotional appeal) and purchasing products significantly reduced their subsequent personal environmental behavior intention, while viewing non-green advertising and purchasing products had no significant effect on their subsequent personal environmental behavior intention; for the high environmental protection, cognition level group stimulus had no effect on their subsequent personal environmental behavior whether they viewed green advertising (emotional appeal) or purchased products from non-green advertising. The above findings answer RQ1, that there is a licensing effect of green advertising in the Chinese consumption context, and H1, H2 and H3 hold.

### 4.2. Environmental Protection Cognition × Advertising Appeal Types (Rational Appeal Green Advertising vs. Non-Green Advertising)

The results of data analysis showed a significant main effect of personal environmental protection cognition (F(1, 116) = 326.827, $p$ = 0.00, Partial $\eta^2$ = 0.738) while the main effect of advertising appeal types (F(1, 116) = 2.076, $p$ = 0.152, Partial $\eta^2$ = 0.018) was not significant. In addition, there was no significant interaction between personal environmental protection cognition and advertising appeal types on environmental behavioral intention (F(1, 116) = 1.816, $p$ = 0.180, Partial $\eta^2$ = 0.015), as shown in Figure 2.

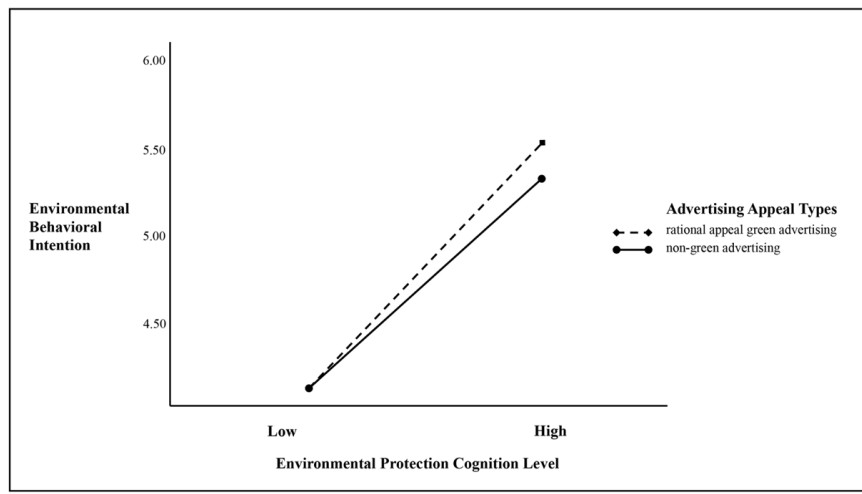

**Figure 2.** Rational Appeal Green Advertising vs. Non-green Advertising.

Further comparative analyses showed that for the low environmental protection cognition level group, there was no significant difference between the personal environmental behavioral intention after viewing rational appeal green advertising and the personal environmental behavioral intention after viewing non-green advertising ($M_{green}$ = 4.133, $SD_{green}$ = 0.299; $M_{non}$ = 4.127, $SD_{non}$ = 0.447; F = 6.013, $p$ = 0.946, 95% CI = [−0.1899; −0.2032]); for the high environmental protection cognition level group, there was no significant difference between the personal environmental behavioral intention after viewing rational appeal green advertising and that of the group viewing non-green advertising ($M_{green}$ = 5.567, $SD_{green}$ = 0.458; $M_{non}$ = 5.327, $SD_{non}$ = 0.499; F = 0.116, $p$ = 0.061, 95% CI = [−0.0094; −0.4094]).

In other words, we can assume that viewing green advertising with rational appeals does not trigger a licensing effect, as verified by H4.

*4.3. Environmental Protection Cognition × Advertising Appeal Types (Rational Appeal Green Advertising vs. Emotional Appeal Green Advertising)*

The results of the data analysis showed a significant main effect of personal environmental protection cognition (F(1, 116) = 656.497, $p$ = 0.00, Partial $\eta^2$ = 0.850), along with a significant main effect of advertising types (F(1, 116) = 4.755, $p$ = 0.031, Partial $\eta^2$ = 0.039). In addition, there was a significant interaction between personal environmental protection cognition and advertisement appeal types on environmental behavioral intention (F(1, 116) = 8.036, $p$ = 0.05, Partial $\eta^2$ = 0.065), as shown in Figure 3.

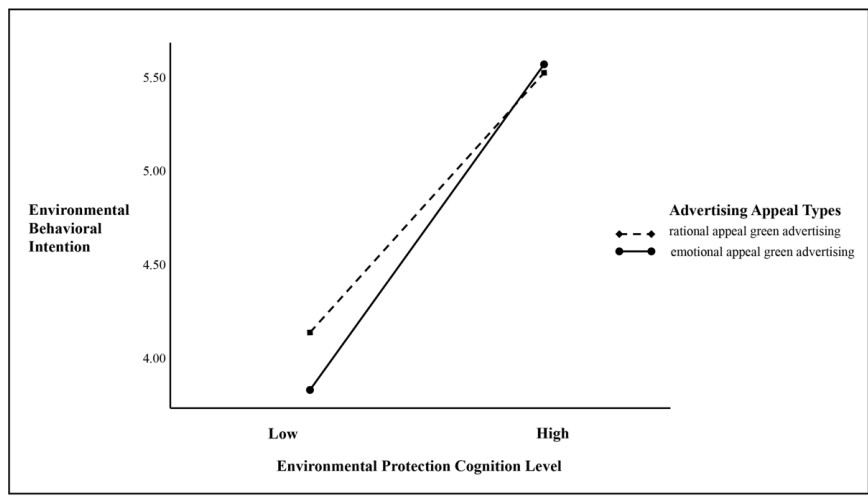

**Figure 3.** Rational Appeal Green Advertising vs. Emotional Appeal Green Advertising.

Further comparative analysis revealed: For the low environmental protection cognition level group, the difference in personal environmental behavioral intention after viewing rational appeal green advertising was significant from that after viewing emotional appeal green advertising ($M_{emotional}$ = 3.827, $SD_{emotional}$ = 0.266; $M_{rational}$ = 4.133, $SD_{rational}$ = 0.299; F = 0.029, $p$ = 0.000, 95% CI = [−0.4529; −0.1603]). For the high environmental protection cognition level group, there was no significant difference between personal environmental behavioral intention after viewing rational appeal green advertising and personal environmental behavioral intention after viewing emotional appeal green advertising group ($M_{emotional}$ = 5.567, $SD_{emotional}$ = 0.458; $M_{rational}$ = 5.527, $SD_{rational}$ = 0.280; F = 9.365, $p$ = 0.685, 95% CI = [−0.1563; 0.2363]). Generally, we can conclude that rational green advertising has a better effect on personal environmental behavioral intention than emotional appeal green advertising, especially for the low environmental protection cognition level group, and the effect is significant, as verified by H5.

## 5. Discussion

*5.1. Licensing Effect Is Universal All over the World*

Previous studies have proved that licensing effect exists in many countries in America and Europe [44]. Although previous studies have pointed out that different cultural backgrounds may lead to differences in licensing effect [45], the conclusion of this study proves that licensing effect in green advertising is also applicable to China's green consumption situation. Therefore, we guess that licensing effect may be a global problem and appear in countries with different cultural backgrounds. We also call on scholars from different countries in the world to further supplement and verify it in their own consumption situations. China is one of the most populous countries in the world, and it is also one of the most important consumer markets in the world. Therefore, it is very important to explore the green consumption in China for global environmental protection [46]. The licensing effect exists in China's consumption situation,

which should arouse the attention of scholars. We call on scholars to further reveal the causes of licensing effect and the ways to alleviate this effect.

### 5.2. Individual Environmental Protection Cognition as the Inducement of Licensing Effect

Previous studies on the licensing effect have revealed some factors that may cause the licensing effect, such as goal satisfaction, situational explanation, compensatory motivation, etc. [47,48]. In this study, individual environmental protection cognition is introduced as the trigger factor of licensing effect. Specifically, when individual environmental protection cognition is low, although some green advertisements can effectively induce consumers to buy green products, it also lays a hidden danger for persistent non-green behaviors. From another point of view, on the surface, the licensing effect is contrary to the cognitive congruence theory [49], while from the perspective of individual environmental protection cognition, the follow-up non-green consumption is actually consistent with the low individual environmental protection cognition. Therefore, in fact, the licensing effect is a manifestation of cognitive congruence theory. This paper only explores the licensing effect from the dimension of individual environmental protection cognition, so it calls on future researchers to explore a wider range, such as individuals focusing on goal progress [50].

### 5.3. Green Advertising with Rational Appeal, an Effective Alleviative of Licensing Effect

In the research on advertising appeal, rational appeal is considered to be effective in changing consumers' cognition from a deep level [51]. Therefore, this study tries to compare green advertisements with rational appeal and emotional appeal. The result shows that the green advertisement of rational appeal does not cause licensing effect, while the green advertisement of emotional appeal does. Therefore, it can be figured out that adopting rational green advertising in green advertising marketing practices can prevent consumers with low environmental protection cognition from taking retaliatory non-green behaviors in the future. Therefore, we encourage advertisers to try their best to adopt rational green advertisements in green marketing, so as to fundamentally improve people's awareness of environmental protection. We also call for future research on green advertising to be cut in from other angles, such as different green advertising appeals (self-benefit vs. other-benefit appeals) [52], in order to find more green advertising methods to reduce the licensing effect.

To sum up, from our experimental results, we have made the following summary: (1) The licensing effect is also applicable to the Chinese consumption context; (2) The level of individual environmental protection cognition is significantly related to individual environmental behavioral intention; (3) The licensing effect is not induced when audiences watch non-green advertisements and rational appeals green advertisements; however, when the green advertisements are emotional appeals, the licensing effect only occurs in individuals with low environmental protection cognition level; there is no licensing effect in individuals with high environmental protection cognition level. In summary, our research hypotheses were all verified.

### 5.4. Implication

From the theoretical perspective, firstly, this study complements the existing research in the fields of green marketing, green advertising, and consumer behavior. The current study enriches the research scope of green advertising consumer behavior. The main focus of existing research on the effects of green advertising has mostly been driven in the direction of green advertising attitudes, green consumer behavior intentions or behaviors, and advertising influence factors [53–55]. However, the ultimate goal of green advertising should be to promote people's sustainable green behavior, and this part of the study has been neglected for a long time. Therefore, future research can also focus on other subsequent green behaviors of the audience, such as consumer green engagement, green value co-creation, and other behaviors that take advantage of individual initiative. Second, this study complements the applicable scope of research on the licensing effect of international green advertising and verifies the existence of the licensing effect in the

Chinese consumption context; however, there is also a high degree of complexity and variability within the Chinese consumption context. Studies have shown that different geographic spaces and different cultural differences in China may be influential factors in audience behavior, and future research can refine this in greater depth. Finally, this study not only re-verifies the influence of consumer traits on green consumption and green behavior through the perspective of the individual environmental cognitive level but also explains the apparent paradox between the licensing effect and cognitive dissonance theory by comparing different green advertising appeals and explores one way to reduce the licensing effect and expand the scope of green advertising research. However, it has also been shown that factors such as consumer psychological distance [56] and advertising target frame [57] can also have an impact on advertising effectiveness and audience behavior, and these factors may also be other explanatory variables for the licensing effect, so they can be taken into consideration and explored in depth in future studies to better achieve the ultimate goal of green advertising.

From the practical perspective, this study provides guidance to governments and companies for better green advertising communication as well as green marketing practices, as our study proves that green advertising does not always lead to sustainable green behavior. Given that the level of individual environmental protection cognition is an important factor in the generation of the licensing effect, working to improve the environmental protection cognition and awareness of the public at the overall social level is an important prerequisite for improving the effectiveness of green advertising in the long term. In addition, given the effective moderating effect of advertising appeals, the planning of green advertising should avoid too many emotional factors, and should instead focus on improving the scientific and professional content of advertising.

### 5.5. Limitations

This study also has some limitations. First, the study was conducted based on an experimental context, which differs from the real advertising environment, which may affect the generalizability of our findings. Second, the experimental stimuli in this study were daily products, ignoring the richness of the products, and whether the findings are also applicable to other product categories remains to be added in future studies. Third, we did not take demographic characteristics such as education level and income level into account, which could be attempted in future studies. Finally, our H1 may have been verified by some literature, but our research is based on the consumption situation in China, so we have repeatedly verified it.

### 6. Conclusions

Green advertising has become one of the most important forms of advertising worldwide, and the licensing effect is already a global problem. Our findings underscore the importance of considering post-purchase licensing effects when brands use green advertising, as consumers may feel entitled to reduce their environmental concerns and behavioral intentions to protect the environment after purchasing the products advertised in green advertising. Yet, these effects typically occur only for consumers with lower environmental protection cognition levels; consumers with higher environmental protection cognition levels do not exhibit these licensing effects. Thus, licensing effects may undermine long-term environmentally friendly consumption. However, green advertising rational appeals can effectively eliminate the licensing effect, thus scientific and rigorous rational appeals should be used as much as possible in green advertising communication.

**Author Contributions:** Conceptualization, C.G.; methodology, C.G.; software, C.G.; validation, C.G.; formal analysis, C.G.; investigation, C.G.; resources, C.G.; data curation, C.G.; writing—original draft preparation, C.G.; writing—review and editing, C.G.; visualization, C.G.; language touch-ups, S.L.; supervision, S.C.; project administration, S.C.; funding acquisition, C.G. All authors have read and agreed to the published version of the manuscript.

**Funding:** This research was supported by the Shenzhen Philosophy and Social Science 2020 Project " Research on the Construction Path of Shenzhen City Culture Brand Symbol System under the Perspective of Digital Intelligence Communication" (SZ2020B034).

**Institutional Review Board Statement:** This study was approved by the academic committee of the school of Journalism and communication of Xiamen University (protocol code: 20220520), and we carefully verified that we complied strictly with the ethical guidelines.

**Informed Consent Statement:** Informed consent was obtained from all subjects involved in the study.

**Data Availability Statement:** The data presented in this study are available upon request from the authors.

**Acknowledgments:** The authors thank all the participants of this study. The participants were all informed about the purpose and content of the study and voluntarily agreed to participate. The participants were able to stop participating at any time without penalty.

**Conflicts of Interest:** The authors declare no conflict of interest.

## Appendix A

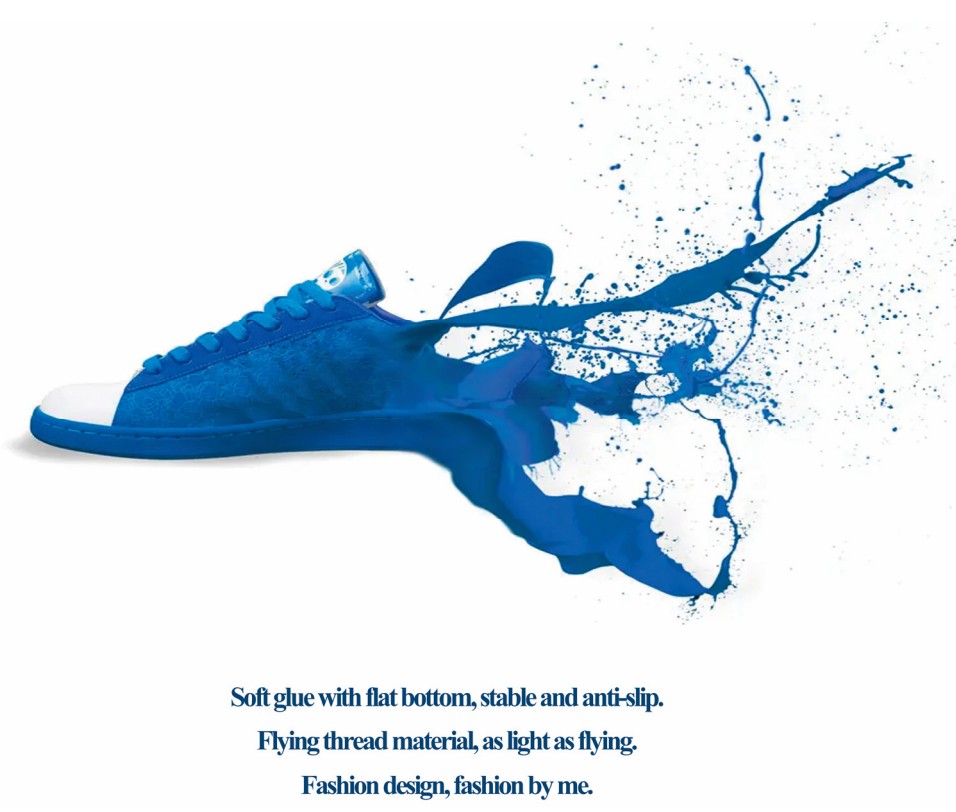

<div align="center">

**Soft glue with flat bottom, stable and anti-slip.**

**Flying thread material, as light as flying.**

**Fashion design, fashion by me.**

</div>

**Figure A1.** Non-green Advertising.

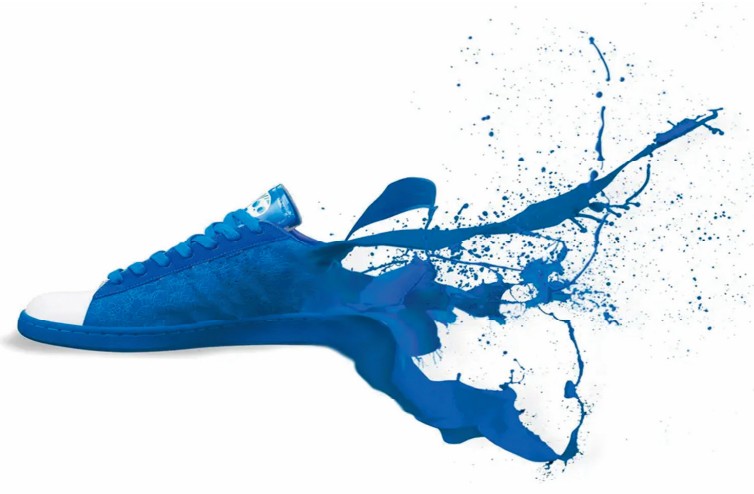

All shoes are made of environmentally friendly materials, which will bring you a green experi-ence. Environmental protection materials are friendlier to the environment, making the water greener and the soil cleaner. Your choice adds green to the natural environment and contributes to protecting the ecological home on which human beings depend.

**Figure A2.** Emotional Appeal Green Advertising.

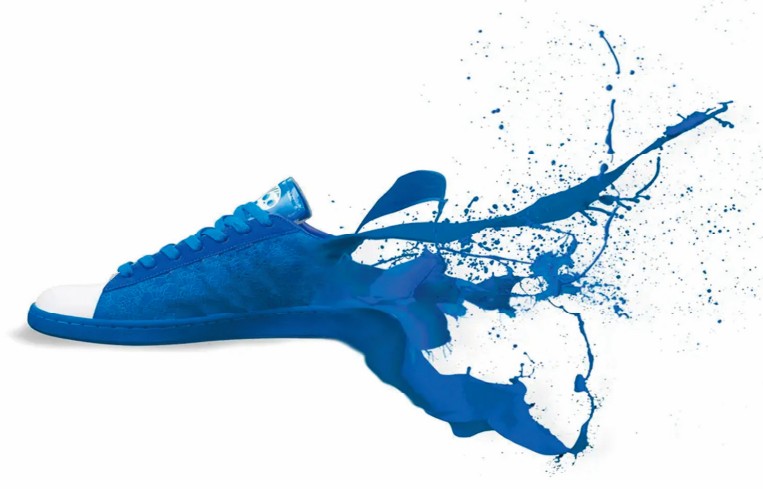

The carbon emission of producing a pair of traditional sports shoes is 14 kg. This shoe adopts brand-new PRIMEGREEN environmentally friendly yarn, and the TPEE rubber sole can be recy-cled, which ensures the function and can be effectively recycled. Your choice can reduce the carbon emission generated by traditional sports shoes by 95%. TIPS: Studies show that excessive carbon emissions are related to global greenhouse effect, melting of glaciers in the South and North Poles, and global extreme climate.

**Figure A3.** Rational Appeal Green Advertising.

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
