# Peer review of "Does Viewing Green Advertising Promote Sustainable Environmental Behavior? An Experimental Study of the Licensing Effect of Green Advertising"

_sustainability, doi:10.3390/su142215100_

Round 1

Reviewer 1 Report

Dear authors, 

In order to improve your manuscript I consider the following issues:

1.the manuscript is poor referenced so you need to consider to take a deeper research

2.the discussion section is very short. This section needs a much more attention from you, to improve it and comment them by refering to other researches. 

Author Response

Response to Reviewer 1 Comments

Manuscript ID: sustainability-1923254

Dear reviewer 1:

Thank you for this valuable feedback. Your comments were highly insightful and enabled us to greatly improve the quality of our manuscript. In the following pages are our point-by-point responses to each of yours comments:

Point 1: The manuscript is poor referenced so you need to consider to take a deeper research.

Response 1: Thank you for your valuable suggestions, and we agree with your opinions. Therefore, this paper revised the literature review and supplemented the references of related research in recent years to highlight the research value and logic of this paper (Changes of references were marked yellow).

Point 2: The discussion section is very short. This section needs a much more attention from you, to improve it and comment them by refering to other researches.

Response 2: Thanks for your valuable opinion, which is very important to improve the quality of our manuscripts. We have rewritten the discussion part and explained it from three aspects: 1. China has a huge population base and is also the most important consumer market in the world. Therefore, it is necessary to discuss the licensing effect in the consumption situation of China. 2. Considering individual environmental awareness as the main cause of permit effect, and discussing about cognitive imbalance theory. 3. Discussing about the practical value of different forms of advertising appeals. (Line 351-402)

Other revisions to the manuscript: In order to improve the English presentation of our manuscript, we invited an English professional author to touch up the language of the article (Large changes in sentences were marked yellow). We hope that the revised manuscript will improve your review experience.

Thank you very much for carefully reviewing our manuscript. Your comments are very useful for the improvement of our manuscript, and we also hope our answers are clear enough. If there is anything need to be improved, please feel free to contact us at any time. Appreciate your work again.

Reviewer 2 Report

The topic is truly interesting, and the research gap is well defined. References should be newer, if possible, also from the last 3-5 years. Please add 1-2 references from Sustainability too. Please try to cite papers which are newer. Old papers from the last 10-20 years do not necessary contain the newest debates from the literature. It is therefore recommended that the newest papers are cited.

In literature review it should be clarified if green advertising of a specific brand or green advertising to promote sustainability without persuading to buy a specific brand was intended to use.

I think H1 has already been confirmed by others.  If someone is a truly convinced environmentalist, he or she is obviously less likely to develop cognitive dissonance than a person with lower level of environmental beliefs.

There is an unfinished sentence in 188-189 lines.

Please clarify sample composition of the samples. 

In chapter 3.2 Manipulation might not be the right word.

Please add the ads that you used in the experiment to an appendix!

Please indicate the unit of measurement in the tables!

In line 383, there is a typing error: “5.2. Limitaions “

The references section must be re-formatted using stability/MDPI format.

Author Response

Response to Reviewer 2 Comments

Manuscript ID: sustainability-1923254

Dear reviewer 2:

We sincerely thank you for your professional review work on our manuscript. As you are concerned, there are several problems that need to be addressed. According to your constructive suggestions, we have made extensive corrections to our previous draft, the detailed corrections are listed below:

Point 1: The topic is truly interesting, and the research gap is well defined. References should be newer, if possible, also from the last 3-5 years. Please add 1-2 references from Sustainability too. Please try to cite papers which are newer. Old papers from the last 10-20 years do not necessary contain the newest debates from the literature. It is therefore recommended that the newest papers are cited.

Response 1: Thank you for your valuable advice. We have added references to Sustainability, and at the same time, we have supplemented some papers in recent years and replaced the papers in the past 10-20 years (Changes of references were marked yellow). Thank you for your opinion.

Point 2: In literature review it should be clarified if green advertising of a specific brand or green advertising to promote sustainability without persuading to buy a specific brand was intended to use.

Response 2: We very much appreciate your opinion, which allows us to improve the quality of our manuscripts. In the literature review, we stated that the green advertisement used in this paper has the following characteristics: 1. No specific brand is used in the advertisement to prevent the influence of the brand's existing attitude on behavior. (We supplemented the advertising pictures used in the experiment in the appendix, and explained the necessity of not using specific brands through references.) 2. The green advertising we adopted is to protect the environment by calling on people to buy products made of environmentally friendly materials.

Point 3: I think H1 has already been confirmed by others.  If someone is a truly convinced environmentalist, he or she is obviously less likely to develop cognitive dissonance than a person with lower level of environmental beliefs.

Response 3: Thank you for your suggestion, and we appreciate it a lot. H1 has indeed been confirmed by some literatures. However, we still beg to keep H1, because it is a study on the licensing effect of green advertising in the certain consumption situation of China. Therefore, although it is a common sense verification, it may be feasible to verify the differences in different cultures again. We fully respect your opinion, so we have made necessary explanations in the limitation section. Thank you again for your seriousness.

Point 4:

4.1 There is an unfinished sentence in 188-189 lines.

Response: This sentence is unnecessary, so we deleted it. Thank you for your careful review. We are sorry for our mistakes.

4.2 Please clarify sample composition of the samples.

Response: We supplemented the specific composition of the sample (table 1).

4.3 In chapter 3.2 Manipulation might not be the right word.

Response: Thank you for your opinion. We replaced ‘Manipulation of Advertising Appeal Types’ with ‘Advertising Appeal Types as Variable’.

4.4 Please add the ads that you used in the experiment to an appendix!

Response: In the appendix section, we supplemented the advertisement for our experiment and described the copywriting content.

4.5 Please indicate the unit of measurement in the tables!

Response: Thank you for your opinion. We marked the unit of measurement in the table.

4.6 In line 383, there is a typing error: “5.2. Limitaions “

Response: Thank you for your review. We corrected this mistake.

4.7 The references section must be re-formatted using stability/MDPI format.

Response 4: Thank you for your opinion. We have corrected the format of our references.

Other revisions to the manuscript: In order to improve the English presentation of our manuscript, we invited an English professional author to touch up the language of the article (Large changes in sentences were marked yellow). We hope that the revised manuscript will improve your review experience.

Thank you again for your professional review work on our manuscript. We hope that changes we have made could resolve all your concerns about the article. I’m more than happy to make any further changes that will improve the manuscript and facilitate successful publication.

Reviewer 3 Report

The manuscript seems to add a valuable contribution to an important discussion and the suggestions made in the conclusions are useful. The research idea is easy to follow, with abundant references to published works. The analyses seem well-founded and accurate.

Typos

18: there seems to be a typo, I think the subject of the sentence is missing

23: Full stop after advertising

62: [Y]et

108-110: it is not clear to me what is meant

135-136: cognitive coherence theory/cognitive congruence theory

152: could be more appropriate: themselves?

188-189: I don’t understand the meaning of the sentence

383: limitations

Author Response

Response to Reviewer 3 Comments

Manuscript ID: sustainability-1923254

Dear reviewer 3:

Thank you for this valuable feedback. Your comments were highly insightful and enabled us to greatly improve the quality of our manuscript. In the following pages are our point-by-point responses to each of yours comments:

Point :  18: there seems to be a typo, I think the subject of the sentence is missing

23: Full stop after advertising

62: [Y]et

108-110: it is not clear to me what is meant

135-136: cognitive coherence theory/cognitive congruence theory

152: could be more appropriate: themselves?

188-189: I don’t understand the meaning of the sentence

383: limitations.

Response: Thank you for your careful review, we have corrected these mistakes.

Other revisions to the manuscript: In order to improve the English presentation of our manuscript, we invited an English professional author to touch up the language of the article (Large changes in sentences were marked yellow). We hope that the revised manuscript will improve your review experience.

Thank you very much for carefully reviewing our manuscript. Your comments are very useful for the improvement of our manuscript, and we also hope our answers are clear enough. If there is anything need to be improved, please feel free to contact us at any time. Appreciate your work again.

Round 2

Reviewer 1 Report

Dear authors, 

It is better now that you improved your paper